# Factors Influencing Cassava Sales and Income Generation among Cassava Producers in South Africa

**Bernard Manganyi, Moses Herbert Lubinga \*** **, Bhekani Zondo and Ndiadivha Tempia**

Markets and Economic Research Centre, National Agricultural Marketing Council, Private Bag X935, Pretoria 0001, South Africa; bmanganyi@namc.co.za (B.M.)

\* Correspondence: moseslubinga@yahoo.co.uk

**Abstract:** Assessing the factors influencing cassava sales and income generation among South African cassava farmers is critical for informed decision-making, policy formulation, targeted interventions, and the long-term growth of the cassava value chain. By recognizing these elements, stakeholders can improve market efficiency, increase income opportunities, reduce poverty, promote rural development, and nurture a sustainable and inclusive cassava value chain. This study examines factors that influence cassava sales and positive income generation along the cassava value chain. Using a simple sample method, we collected data from 240 farmers in the South African provinces of KwaZulu-Natal, Limpopo, and Mpumalanga. A logistic regression was used to investigate the impact of explanatory variables on the probability of selling cassava and earning a positive income. Findings show that having access to output markets, owning livestock, being a female, and having sizable land under cassava production enhances the possibility of cassava sales and generating a positive income. In contrast, age has a detrimental influence on cassava sales, while access to extension services and harvesting for household food consumption exhibited no substantial effects. The findings underline the importance of market access, gender equality, integrating livestock farming among cassava farmers, support for elderly farmers, and sustainable agricultural practices. To ensure the long-term positive generation of income by farmers and the sustainability of the cassava value chain, policymakers and stakeholders must collaborate and execute policies and interventions that address these essential concerns.

**Keywords:** cassava value chain; gender equity; income generation; logit regression; market access

## 1. Introduction

Cassava, a tropical plant, originated in South America and is now found in many parts of the world, particularly in subtropical and tropical climates [1]. As a result, cassava has become a staple food for more than 800 million people worldwide [2]. According to [3], Nigeria was the world's leading cassava producer in 2021 with an average of 63.03 million metric tons (MT) produced, followed by the Democratic Republic of Congo (45.67 million MT), Thailand (30.11 million MT), Ghana (22.68 million MT), and Brazil (18.1 million MT), among others. In South Africa, cassava is grown by smallholder farmers as a secondary crop and is used to produce starch, among other uses [4,5]. It is cultivated in Limpopo, Mpumalanga, and northern KwaZulu-Natal provinces, with significant production areas in Limpopo [1,5].

Cassava is well-known for its adaptability to various agroecological conditions and its capacity to grow in harsh environmental conditions [6,7]. One of the reasons why cassava is seen as beneficial for smallholder farmers is the fact that, in comparison to other crops, it has a comparatively low cost of inputs since its cultivation requires only a small initial expenditure in terms of land preparation and the construction of infrastructure. Cultivating it in various soil types, including marginal and less fertile soils, reduces the amount of agro-inputs required. Stem cuttings are commonly used to reproduce it, thereby eliminating

the need to purchase more expensive seeds [5]. In the absence of pests and diseases, like the Cassava Mosaic Disease (CMD), farmers can reuse their planting materials, thereby lowering the overall cost of procuring fresh seeds for each new planting season.

Beyond being a staple food crop globally (mainly for rural impoverished households), cassava poses a great potential for generating income through the sale of fresh cassava tubers and other processed derivatives. According to [8], in several countries such as Nigeria, Brazil, and Thailand, value addition has shown immense potential to enable cassava growers, processors, and/or traders to earn premium prices within the cassava value chain as opposed to selling the perishable tubers. Furthermore, cassava's ability to yield significant edible biomass per unit of land can potentially increase income. According to [9], income generated through cassava production also plays a highly critical role in the household welfare of farmers. On the other hand, the breadth of cassava applications in industries such as food, animal feed, and pharmaceuticals also boosts farmers' market demand and revenue prospects [5,6,10]. Cassava production and processing provide employment possibilities for rural economic growth and poverty alleviation [5,11,12].

Despite all the abovementioned attributes, cassava production and its marketing in South Africa is by far much lower than the other traditional starch crops such as maize, potatoes, and wheat. The current low level of cassava production in South Africa and the fragmented marketing mechanisms are attributed to several factors, such as market access constraints and an undeveloped value chain in general [5]. From a market access perspective, refs. [13,14] posit that a higher cassava productivity can increase the quantity available for sale, thereby leading to higher sales by volume and potentially a higher accrued income to farmers. At the production stage, farm size and the extent of production are significant determinants of the level of commercialization, which is a good measure of sales and income. Commercialization refers to farmers participating in the market and selling their cassava produce [15]. Ref. [16] examined the impact of cassava commercialization on household income in Kilifi County (Kenya) and commercialization was found to have a significant positive impact on income.

At the same time, commercialization is strengthened by access to markets, which can significantly improve sales and income. Farmers with better access to markets and value chains can benefit from higher prices and increased demand for their cassava produce. Factors such as proximity to markets, transportation infrastructure, and market information can affect farmers' ability to access markets and value chains [17–19]. For the socioeconomic factors, cassava sales and income can be influenced by labour, education, gender, household size, age, and access to financial services. Cassava is a labour-intensive crop, thus using new agricultural machinery can help reduce the time and labour required to cultivate cassava, thereby freeing farmers' time to focus on other activities that can improve their income [20]. Using a probit model, ref. [21] assessed the determinants of participation decisions in cassava marketing by smallholder farmers in Taita-Taveta and Kilifi Counties, Kenya. Their findings suggest that individual factors, such as the farmer's gender, level of education, and household size, influence farmers' decisions to participate in cassava marketing. Refs. [22,23] report that demographic variables such as age, household size, years spent in school, and gender can influence income levels and profitability in cassava production.

Although there is an increasing body of literature that articulates the various contributing factors to cassava sales and the income generated from the cassava value chain, none of the studies focus on South Africa. Therefore, this study assessed the factors influencing farm sales and positive income derived from cassava. The rationale of this study is to provide a better understanding of factors that influence cassava sales and income generation among producers involved in South Africa's emerging cassava value chain. The value chain in South Africa has unique characteristics that need to be explored. For instance, whereas the value chain is dominated by smallholder farmers, some grow cassava for their own household use while others do so as a business. Moreover, even though pest- and disease-free planting materials exist at the Agricultural Research Council (ARC), South

Africa's premier research institution in the agricultural sector, producers expressed limited access to such materials and as a result, they use other means to access planting materials and later on sell their cassava. Therefore, this paper's contribution to the development and sustainability of the cassava value chain lies in a better understanding of the determinants of cassava sales and positive income generation. To the best of our knowledge, this is the first socioeconomic paper attempting to analyse factors influencing cassava sales and income generated from cassava in a South African context. Insights from this work may be of strategic importance in designing and developing custom-tailored interventions to enable the various actors (producers, traders, and processors, among others) to further develop and sustain the value chain.

Our findings suggest that several factors influence both the probability of selling cassava and the likelihood of generating a positive income. The gender of the farmers, access to output markets, livestock ownership, land size, age, and yield all positively influence cassava sales and the positive income generated from cassava. However, other variables, including access to extension services and harvesting for food consumption at home, do not significantly affect cassava sales and positive income.

## 2. Methodology

### 2.1. Area of Study

The study was conducted within the jurisdiction of Limpopo, Mpumalanga, and KwaZulu-Natal provinces, South Africa as shown in Figure 1. In Limpopo, data were collected from the Mopani district in the local municipalities of Giyani, Tzaneen, Greater Letaba, and Ba-Phalaborwa. However, in KwaZulu-Natal province, data were collected from the uMkhanyakude and King Cetshwayo district municipalities. One principal cassava-producing local municipality was chosen for each district municipality: uMhlathuze for King Cetshwayo and uMhlabuyalingana for uMkhanyakude. For Mpumalanga province, data were collected from farmers in Bushbuckridge local municipality in the Ehlanzeni district.

The rainfall patterns in Limpopo, Mpumalanga, and some parts of KwaZulu-Natal align with the requirements of cassava, making these regions suitable for cultivation. Limpopo is in the northernmost part of South Africa, sharing borders with Botswana, Zimbabwe, and Mozambique. It is known for its diverse landscapes, including the northern bushveld, the central lowveld, and the southern highveld. The climate in Limpopo is subtropical, with hot summers and mild winters. The bushveld region in the province's north has a hot and dry climate, with summer temperatures regularly exceeding 30 degrees Celsius.

Limpopo experiences erratic rainfall, with the province receiving most of its precipitation during the summer months (October to March). Mpumalanga's climate is subtropical, with hot and humid summers and mild winters. The lowveld region in the province's east has higher temperatures and humidity than the highveld region in the west. Summers in Mpumalanga can be wet and stormy. Winters are usually dry and mild, with temperatures between 10 and 25 degrees Celsius. KwaZulu-Natal has a diverse climate; coastal areas have a subtropical climate with hot and humid summers, mild winters, and abundant yearly rainfall.

### 2.2. Data Collection

This study used a semi-structured questionnaire to collect data from farmers in the study area. The questionnaire was designed to collect data on various aspects of the cassava value chain, such as cultivation practices, processing, marketing, and income generation. Before beginning data collection, the necessary ethical considerations were made. This process included obtaining informed consent from the farmers, ensuring their privacy and confidentiality, and explaining the purpose of the study. The participants were chosen using a simple random sampling method. As a result, this study randomly sampled 240 farmers from the population of cassava farmers in the study area. This sampling method ensures that each farmer in the population has an equal chance of being included in the study,

reducing bias and increasing sample representativeness. Following the data collection phase, the questionnaire responses were manually entered into a spreadsheet for further analysis. Data management processes such as data cleaning and coding were used to ensure the accuracy and consistency of the data.

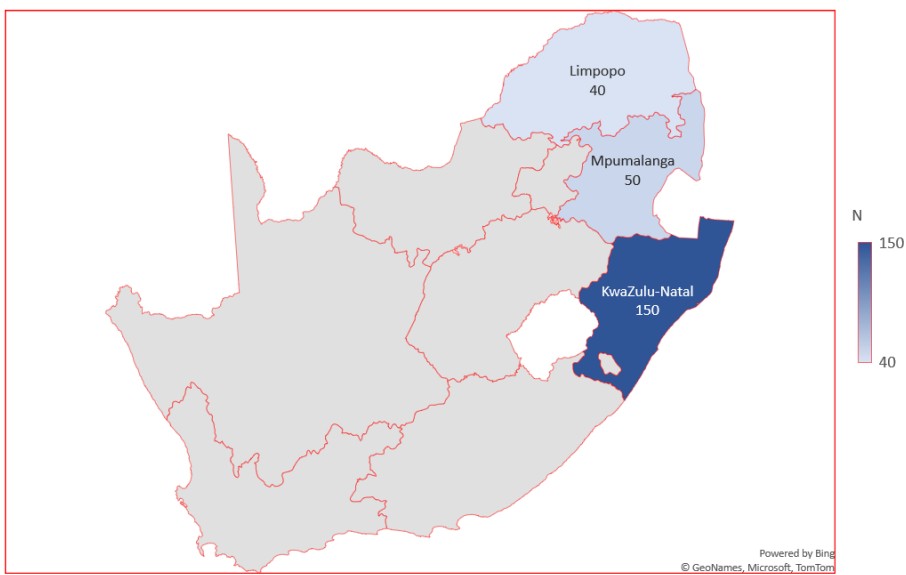

**Figure 1.** Area of study.

*2.3. Empirical Framework*

We used the logit model to examine the link between various explanatory variables and cassava sales. Based on an essential logistic random variable's cumulative distribution function (CDF), the logit model provides a probabilistic technique for estimating the likelihood of factors influencing income outcomes. Logistic regression is a widely used method for binary and multi-class classification tasks. It is a statistical model that assumes a linear relationship between the independent variables and the outcome variable [24]. Logistic regression has been applied in agriculture to examine the impact of factors such as cropland abandonment, direct marketing strategies, market participation, technology adoption, climate-smart agricultural practices, and market accessibility on income in agriculture [25]. Logistic regression provides a valuable tool for understanding the relationships between these factors and their influence on market access and income in the agricultural sector [25–27]. Amid the advantages of logistic regression, it has some limitations. One of the limitations of logistic regression is its assumption of a linear relationship between the independent variables and the outcome. In reality, the relationship may not always be linear, and logistic regression may not capture complex data relationships [28]. This limitation can be addressed by using more flexible models such as generalized additive models (GAMs). Outliers can have a significant impact on the estimated coefficients and distort the results [29]. Robust logistic regression methods have been developed to address this issue and provide more reliable estimates in the presence of outliers [30]. In this study, the logit model expresses the probability ($P_i$) of observing high-income outcomes as a function of the independent variables ($X_i$) and their corresponding coefficients ($\beta$). The formulation is given by:

$$P_i = \Lambda(X \cdot \beta) = \frac{\exp(X \cdot \beta)}{(1 + \exp(X \cdot \beta))}$$

$$\text{logit}(Pi) = \log\left(\frac{Pi}{1 - Pi}\right) = X \cdot \beta$$

(1)

We observe (binary) response variable $Y_i$, which is linked to an unobservable (latent) variable $Y_i^*$ as follows:

$$\left\{ \begin{array}{l} 1, \text{ if } Y_i^* > 0 \\ 0, \text{ if } Y_i^* i \leq 0 \end{array} \right\} \tag{2}$$

In this case, $Y_i^*$ is unobservable, but we hypothesize that it is a linear function of the independent variables:

$$Y_i^* = \beta_0 + \beta_1 X_{1i} + \beta_2 X_{2i} + \cdots + \beta_n X_{ni} + \varepsilon_i = X_i \beta + \varepsilon_i \tag{3}$$

Therefore, we assume that $\varepsilon$ is independent of $X$ and has a known distribution. Regarding the logit model, $\varepsilon$ has a standard logistic distribution.

In this equation, $\beta_0$ represents the intercept or constant term, $\beta_s$ represents the coefficients associated with each independent variable, and $\varepsilon$ represents the error term. In terms of the probit model, $\varepsilon$ has a standard normal distribution. To estimate the unknown parameters ($\beta$) such that the probability of observing the $Y_i$ is as high as possible, we use maximum likelihood. Therefore, we assume that

$$E(y_i \mid X_i) = G(\beta_0 + \beta_0 X_i) = G(X\beta) \tag{4}$$

where $X_i$ and $y_i$ are observed data, and $\beta$s are unknown. In this case, the likelihood function is given as follows:

$$L(\beta) = \prod_{i=1}^{n} [G(X_i \beta)]^{y_i} [1 - G(X_i \beta)]^{1-y_i} \tag{5}$$

The maximum likelihood estimate (MLE) of $\beta$, denoted by $\beta^{**}$, is the $\beta$ that maximizes the log-likelihood function. The general theory of MLE for random samples implies that under very general conditions, the MLE is consistent, asymptotically standard, and efficient.

Therefore, we applied logit regression to comprehend the determinants of cassava sales and the generation of positive income. Logistic regression is particularly well-suited when the outcome variable is binary or categorical, as is the case with our study. In this study, we want to determine whether certain factors increase the likelihood of cassava sales or income generation, making logistic regression an appropriate choice. The underlying motivation to study factors influencing cassava sales and positive income is the need to better understand the economic well-being of cassava farmers. This allows for a well-rounded economic assessment, encompassing both the quantity of cassava sold and the financial outcome after expenses, recognizing that high sales do not guarantee positive income. Therefore, identifying factors influencing cassava sales and positive income can help in efficient resource allocation, enabling targeted efforts for enhanced profitability. This can inform policy formulation, allowing for customized interventions that support small-scale cassava farmers and ensure the long-term economic sustainability of cassava farming.

The logistic models are specified as follows:

Model 1:

$$\begin{aligned} \text{Cassava Sales} = \beta_0 &+ \beta_1(\text{Yield}) + \beta_2(\text{Gender}) + \beta_3(\text{Access to output market}) \\ &+ \beta_4(\text{Own livestock}) + \beta_5(\text{Access to extension}) + \beta_6(\text{Land size}) \\ &+ \beta_7(\text{Harvesting aspect for food}) + \beta_8(\text{Age}) + \varepsilon \end{aligned}$$

Model 2:

$$\begin{aligned} \text{Positive Income} = \beta_0 &+ \beta_1(\text{Yield}) + \beta_2(\text{Gender}) + \beta_3(\text{Access to output market}) \\ &+ \beta_4(\text{Own livestock}) + \beta_5(\text{Access to extension}) + \beta_6(\text{Land size}) \\ &+ \beta_7(\text{Harvesting aspect food}) + \beta_8(\text{Age}) + \varepsilon \end{aligned}$$

Cassava sale is a binary-dependent variable, which represents whether a farmer was engaged in the sale of cassava during the previous harvesting season. A value of 1 is assigned if the farmer sold cassava during that period, while a value of 0 is assigned if no sales were made. The positive income derived from cassava indicates whether a farmer generated a positive net income from their cassava farming activities after deducting all associated costs. In this context, positive income refers to income that exceeds the total expenses related to cassava production. If a farmer's income from cassava exceeded their production costs and resulted in a net profit greater than zero, a value of 1 is assigned. Conversely, if the income did not cover the costs, resulting in a net loss or zero profit, a value of 0 is assigned.

For the cassava sales and positive income models, we test the following hypotheses. Null hypothesis (H0): Independent variables (yield, gender, access to output market, own livestock, access to extension, land size, harvesting aspect food, age) do not have a significant influence on the dependent variables (cassava sales and positive income). Alternative hypothesis (H1): At least one of the independent variables exerts a significant influence on cassava sales and positive income.

Expected signs of each parameter estimate of the explanatory variables for both Model 1 and Model 2 are described in accordance with the economic theory. For yield ($\beta_1$), it is anticipated that an increase in cassava yield leads to higher cassava sales and a positive income since a higher yield implies a greater quantity of cassava available for sale. The expected sign of the estimate for gender ($\beta_2$) in both models is indeterminate due to potential differences in access to resources or market opportunities, coupled with the specific gender dynamics within the agricultural sector. The coefficient on access to output market ($\beta_3$) is expected to be positive for both models (cassava sales and positive income) since improved market access translates to better opportunities to make sales and the generation of positive income. Positive coefficients are expected for own livestock ($\beta_4$). Livestock is associated with an additional source of income that might be used to enhance cassava production, hence higher cassava sales and positive income generated. Access to extension ($\beta_5$) services is anticipated to have a positive relationship with cassava sales and a generation of positive income, as it can enhance farming knowledge and practices. A larger land size allocated for cassava ($\beta_6$) is expected to increase the potential for cassava sales given that more land implies a higher production capacity. Specific practices related to the harvesting aspect for food ($\beta_7$) may negatively affect the quantity of cassava available for sale due to household consumption. Lastly, age ($\beta_8$) is expected to be associated with experience and knowledge, potentially exerting a positive influence on cassava sales and positive income since older farmers may possess more established networks and expertise.

For the positive income model, a similar hypothesis was tested as performed for the cassava sales model. The expected economic theory implications for the variables in Model 2 mirror those in Model 1, albeit with a shift in focus from cassava sales to positive income. Consequently, the same expectations regarding the direction of effects for each variable apply within this context.

## 3. Results and Analysis

### 3.1. Descriptive Statistics of Selected Variables

Table 1 shows the descriptive statistics of respondent farmers. Approximately 68% of the respondents were female. This result suggests that ignoring gender distribution will overlook critical constraints, opportunities, and impacts in the cassava value chain. Gender influences resource distribution, decision-making, market access, and output control. Failing to consider gender can hinder agricultural transformation, productivity, poverty reduction, and improvement in well-being. Addressing gender disparities is essential for impactful interventions in agriculture [31]. Approximately 36% of the individuals in the sample were married. This percentage suggests that a significant portion of the respondents were married. Marriage often signifies a committed partnership between

individuals, and family dynamics and responsibilities may influence their agricultural activities and decisions.

**Table 1.** Descriptive statistics.

| Variable | Mean | Standard Deviation | Median | Min | Max |
|---|---|---|---|---|---|
| Gender (1 = Male; 0 = Female) | 0.32 | 0.47 | 0.0 | 0 | 1.00 |
| Membership to a farmers' group (1 = Yes; 0 = No) | 0.62 | 0.49 | 1.0 | 0 | 1.00 |
| Years of membership in a farmers' group | 4.49 | 7.80 | 1.0 | 0 | 68.00 |
| Cassava sales (1 = Yes if respondent sold fresh cassava tubers in the previous harvesting season; 0 = No) | 0.62 | 0.49 | 1.0 | 0 | 1.00 |
| Fresh cassava tubers sold in the previous harvesting season (kg) | 169.48 | 378.29 | 1.0 | 0 | 2000 |
| Positive income (1 = Yes if a farmer generated profits from cassava production in the previous season; 0 = No) | 0.68 | 0.52 | 1.0 | 0 | 1.00 |
| Land size (Hectares of land under cassava production) | 0.62 | 1.34 | 0.5 | 0 | 17.13 |
| Own livestock (1 = Yes if respondent owns livestock; 0 = No) | 0.64 | 0.48 | 1.0 | 0 | 1.00 |
| Marital status (1 = Yes if respondent is married; 0 = No, including divorced and single people) | 0.36 | 0.48 | 0.0 | 0 | 1.00 |
| Age of respondent (Years) | 57.52 | 12.70 | 59.0 | 24 | 96.00 |
| Household size (Number of individuals in a household) | 7.00 | 3.51 9 | 7.0 | 1 | 27.00 |
| Years (Period involved in cassava production) | 18.41 | 11.25 | 18.5 | 1 | 40.00 |
| Agric land size (Hectares under agricultural activities) | 2.02 | 4.06 | 1.0 | 0 | 42.00 |
| Farming experience (Years) | 23.27 | 13.33 | 24.0 | 1 | 50.00 |
| Pension (ZAR/month) | 288.12 | 1197.90 | 0.0 | 0 | 15,000 |
| Social grants (ZAR/month) | 677.24 | 840.08 | 0.0 | 0 | 2400 |
| Employment (ZAR/month) | 222.08 | 1187.87 | 0.0 | 0 | 11,000 |
| Yield (Fresh cassava tubers harvested during the previous season—kg/ha) | 368.68 | 661.40 | 150.0 | 0 | 5000 |

About 62% of the individuals in the sample are members of a farmers' group, suggesting that farmers are willing to be organized to amplify their political voice, coordinate market access, create a platform for information sharing, and make collective decisions. Farmers' groups create a conducive environment for farmers to be more involved in value-adding activities such as agro-processing, marketing, distribution, and credit. Farmers' groups provide a platform to unite their voices and negotiate better terms with suppliers, buyers, and government institutions [15]. By collaborating, farmers can achieve more substantial bargaining power and secure fair prices for their produce, favourable input prices, and improved access to markets and services [32]. According to the descriptive statistics, years of membership in a farmers' group is on average 4.49 years, which indicates active participation and the farmer's capacity to use the opportunities provided by the group.

About 62 percent of the people in the sample have sold fresh cassava tubers, and cassava production is not simply performed for the goal of subsistence but also for commercial gains. It shows that a significant proportion of the sampled population is actively involved in the agricultural market, attempting to generate income by selling their cassava produce. On average, respondents sold 169.5 kg of fresh cassava tubers during the previous season, but the maximum quantity sold was 2000 kg. There is a high deviation (378.3) of the quantity of fresh cassava tubers sold given that some producers indicated that they mainly produce for household consumption and occasionally for sale. The average amount of land size allocated to the respondents for cassava production is 0.62 hectares, with an absolute minimum of 0 and an absolute maximum of 17.2 hectares. While the data show that some farmers do not have farming land, the nature of the cassava value chain allows landless farmers to nonetheless participate. For instance, basing on the discussion we had with farmers, it was noted that some farmers source and sell cassava planting materials (cuttings) to other farmers. This finding suggests that cassava cultivation can be undertaken on small land holdings. The study conducted by the authors of [15] discovered that the amount of land available to cassava growers significantly impacts the likelihood of agricultural commercialization. This finding suggests that farmers who have access to more suitable tracts of land are more likely to engage in cassava cultivation and trading on a commercial scale.

In terms of livestock, 64% of the respondents own livestock. Livestock ownership is a part of the broader livelihood strategies of individuals. Livestock rearing complements cassava cultivation and contributes to a more diversified and resilient livelihood portfolio.

For instance, farmers use cassava as feed since livestock, such as ruminants, can efficiently convert the carbohydrates in cassava into energy and protein. Cassava can complement other feed ingredients in balancing the nutritional requirements of livestock, contributing to improved feed efficiency and overall animal performance.

The average age of respondents was 57.52 years old. Because the average age of a cassava farmer is rising, detailed succession planning may soon be required to secure the continuity of agricultural enterprises. Some issues arising from an aging farming population may be mitigated by promoting and supporting younger individuals who are interested in farming, or by developing inter-generational collaborations. The use of new farming technologies and techniques may explain the average age. The average home has seven members. Multiple family members can contribute to greater labour availability, benefiting labour-intensive operations like land preparation, planting, weeding, and harvesting.

Furthermore, a larger household size means a greater demand for food, especially cassava and its derivatives. Cassava farming can help households satisfy their nutritional demands, maintaining food security and potentially lowering their reliance on externally sourced food. Pensions, social grants, and employment represent income levels in South African Rands (ZAR). The average pension is ZAR 288.12, social grants are ZAR 677.24, and job income is ZAR 222.08, which suggest that many cassava farmers experience financial constraints and limited access to significant income sources since the income streams are below the food poverty line (equivalent to ZAR 624 per month) in South Africa. This may limit their ability to invest in agricultural supplies, equipment, and technology that could improve cassava production.

The mean yield for fresh cassava tubers was 368.68 kg/ha, with a minimum of 0 kg/ha and a maximum of 5000 kg/ha. Therefore, the difference between the average and maximum yield recorded (5000 kg/ha) indicates a productivity gap within the sample. This suggests room for improvement in cassava farming practices, as some farmers achieved significantly higher yields. Bridging this gap requires identifying and adopting the best practices and technologies to enhance productivity. With an average farming experience of over two decades (23.3 years), this implies that respondents exhibit accumulated expertise and knowledge in farming in general. Thus, the respondents' long-term exposure to agricultural practices, challenges in farming, and local conditions provide them with essential ideas and abilities that can contribute to enhanced farming techniques, pest and disease management, and overall production.

*3.2. Explaining Factors Influencing Income from Selling Cassava*

3.2.1. Diagnosis

We first assessed invalid probabilities, a practical approach to assess the model's performance. As a result, we observed that the probabilities in the two models are within the expected range (0 to 1), which indicates that the models provide reasonable estimates of the probabilities of farmers selling cassava and generating positive income.

This information is essential because it suggests that the logit captures the relationships between the independent variables and the likelihood or probability of accurately selling cassava and generating positive income. It provides confidence in the validity and reliability of the model results. In addition, we used the Variance Inflation Factor (VIF) to check for multicollinearity. When the independent variables in a regression model are significantly linked with one other, a phenomenon known as multicollinearity occurs. This phenomenon can result in inconsistent and biased estimations of the coefficients. A strong correlation exists between a variable and the other variables in the model if the value of the VIF for that variable is high (usually greater than 10). If the VIF values are low, it indicates that the independent variables in the logistic regression model are not substantially associated with each other. As a result, their impacts on the variable that is being modelled can be approximated more precisely. The fact that the VIF values for each variable in our models are on the low side indicates that there is no major multicollinearity among the independent

variables. Therefore, this suggests that the variables can be utilized entirely in the logistic regression model without adversely impacting high intercorrelations.

Moreover, we conducted the Breusch–Pagan test to examine heteroskedasticity. The results of our investigation show that all models exhibit signs of heteroskedasticity. These findings suggest that the variance of the residuals in the model does not remain the same across all possible levels of the independent variables. Since the logistic and probit regression model is susceptible to producing biased and ineffective coefficient estimates if heteroskedasticity exists, we estimated robust standard errors to cater for heteroskedasticity.

### 3.2.2. Factors Influencing Cassava Sales and Positive Income

We used a logit model to assess the influencing factors of fresh cassava sales and positive income generated from cassava. We ran a normal logit regression and scaled parameters were used to evaluate marginal effects regarding probabilities. Therefore, logit regression coefficients were transformed to log-scaled parameters to estimate the marginal effects in terms of probability instead of representing the probability itself. The coefficients in the logistic regression model are meant to represent the log odds or logit of the likelihood. When we used scaled parameters, we calculated the marginal effects in terms of probabilities, providing a more intuitive explanation of the independent variables' impact on the probability of the result. This transformation enabled us to quantify the change in the likelihood of the outcome variable based on changes in the independent variables, which is frequently more relevant. We used cassava sales and positive income as dependent variables (See Table 1).

### 3.2.3. Cassava Sales

Assessing factors influencing cassava sales provides a better understanding of dynamics associated with the cassava value chain. Farmers' demographics, for example, influence their decision-making processes, resource allocation, and ability to participate successfully in the value chain. Cassava output is heavily influenced by farm-specific factors such as land size, livestock ownership, farming experience, and the presence of other agricultural enterprises. Furthermore, market access is a critical factor influencing farmers' capacity to sell cassava products. In testing our stated null hypothesis (H0) that none of the independent variables (yield, gender, access to output market, own livestock, access to extension, land size, harvesting aspect food, age) has a significant influence on the dependent variable (cassava sales), we found evidence to reject this null hypothesis in favour of the alternative hypothesis (H1) for most of the variables. Table 2 shows that a one-unit increase in yield is associated with a 0.03% increase in the likelihood of selling cassava, which is significant at a 10% level. Being female (as opposed to male) increases the likelihood of selling cassava by 15.2%. This impact is significant at the 5% level. This suggests that female farmers play an essential role in the cassava value chain by actively participating, especially in the primary production phase. This emphasizes the significance of supporting gender equality and women's empowerment throughout the cassava value chain [33].

Findings reveal that having access to output markets enhances the likelihood of selling cassava by 10.6%, which is significant at the 5% level. This strong positive effect highlights the importance of interventions tailored towards improved market access for cassava producers. Policymakers can help farmers maximize the value of their cassava output and improve their livelihoods by enhancing market linkages, providing market information, and removing barriers to market access. Owning animals increases the likelihood of selling cassava by 11.0%, a statistically significant difference at the 5% level. Farmers can benefit from the combination of livestock and cassava farming. For example, cassava processing byproducts, such as cassava peels, can be utilized as livestock feed, lowering feed costs for livestock owners. Similarly, cattle dung can be used as manure during cassava production, thereby increasing yields. Furthermore, animal ownership can improve cassava growers' market access. Cassava producers' market relations can also benefit from livestock

ownership. This finding conforms to a study conducted by the authors of [34] in Ethiopia, where farmers owning more livestock were technically efficient in cassava production.

**Table 2.** Factors influencing cassava sales and positive income.

| | Dependent Variable: | | | |
| | Cassava Sales | Logit Scaled | Positive Cassava Income | Logit Scaled |
|---|---|---|---|---|
| Yield | 0.002 * | 0.0003 | 0.0003 | 0.00004 |
| | (0.001) | | (0.0004) | |
| Gender | 1.101 ** | 0.152 | 1.201 *** | 0.165 |
| | (0.458) | | (0.446) | |
| Access output market | 0.767 ** | 0.106 | 0.938 ** | 0.129 |
| | (0.386) | | (0.367) | |
| Own livestock | 0.831 ** | 0.11 | 0.841 ** | 0.116 |
| | (0.361) | | (0.343) | |
| Access extension | 0.912 | 0.126 | 0.530 | 0.073 |
| | (0.835) | | (0.705) | |
| Land size | 0.113 | 0.0156 | 0.160 * | 0.022 * |
| | (0.093) | | (0.096) | |
| Harvesting aspect food | −0.960 | −0.132 | −0.850 | −0.117 |
| | (0.616) | | (0.556) | |
| Age | −0.028 * | −0.004 | −0.021 | −0.0029 |
| | (0.015) | | (0.014) | |
| Constant | 1.786 * | 0.246 | 1.349 | 0.858 |
| | (1.080) | | (1.004) | |
| Observations | 236 | | 236 | |
| Log likelihood | −99.642 | | −108.621 | |
| Akaike Inf. Crit. | 217.283 | | 235.241 | |

Note: Standard errors are in parentheses. ***, **, and * denote significance levels at 1%, 5%, and 10%, respectively.

Farmers with animals may have established contacts with buyers and market channels due to their livestock sales. This network and market knowledge can facilitate cassava sales, increasing the likelihood of cassava sales. Income diversification can help strengthen household resilience, especially in volatile or risky livestock markets. A one-year increase in age relates to a 0.4% drop in the likelihood of selling cassava, with the effect statistically significant at the 10% level. The labour-intensive operations associated with cassava farming, including planting, harvesting, and processing, may pose difficulties for older farmers. These activities necessitate physical strength and endurance, which in most cases deteriorate as one becomes older. As a result, older farmers may minimize their involvement in cassava sales, affecting cassava supply along the value chain. Older farmers may be less willing to accept new cassava production, marketing technology, and innovations. This can influence their competitiveness and capacity to engage successfully in the value chain. Efforts to assist older farmers in gaining access to and using relevant technologies will help enhance their productivity and enable them to continue selling cassava. The age-related decrease in the likelihood of selling cassava highlights the significance of specific policies and interventions to assist older farmers in the cassava value chain. This could involve tailoring training and capacity-building programmes to their unique requirements, enabling access to other income sources, and addressing specific issues in cassava cultivation and marketing. Ref. [35] explained farmers' income through market orientation and participation and found similar results. The authors discovered that an increase in the age of farmers reduces their productivity.

### 3.2.4. Positive Income

In this case, we tested the null hypothesis (H0) that none of the independent variables (yield, gender, access to output market, own livestock, access to extension, land size, harvesting aspect food, age) has a significant influence on positive income generated from cassava. According to the results (see Table 2), it was evident to reject the null hypothesis in favour of the alternative hypothesis (H1) for most of the variables. Being female, as opposed to being male, increases the probability of generating a positive cassava income by 16.5%; this effect is significant at the 1% level. Therefore, interventions or policies that support

and empower female farmers in the cassava value chain are recommended to enhance income opportunities. The coefficient for gender is 0.165, which suggests that being female increases the probability of a positive cassava income.

Access to output markets has a significant coefficient of 0.129, suggesting that access to output markets increases the probability of a positive income generation from cassava by 12.9%. This highlights the importance of facilitating market access for cassava farmers, as it can directly impact their income opportunities. Policy interventions that enhance market linkages, improve transportation, and provide market information are recommended. The coefficient for the livestock ownership variable is 0.116, which suggests that having livestock raises the likelihood of a positive cassava income by 11.6%, which is significant at a 5% level. This suggests that diversification in agricultural enterprises through livestock ownership can contribute to income stability and resilience among cassava farmers. Encouraging integrated farming practices that combine cassava cultivation with livestock rearing may be beneficial. The land size coefficient is 0.022, which indicates that an increase in one unit of land size under cassava production leads to an increase of 0.022% in the probability of a positive income being generated from cassava, which is significant at a 10% level.

Although coefficients for harvesting food and access to the extension services were positive yet age was negative, they were statistically insignificant. This implies that other factors not included in the model, or a larger sample size, may be needed to better understand the relationship between each variable and income generated from cassava. In the case of access to extension services, it is essential to consider the quality and effectiveness of extension services rendered to cassava farmers while further exploring their influence on improving positive income generated along the cassava value chain.

## 4. Conclusions and Recommendations

This study investigated factors that influence cassava sales and positive income generated from the cassava value chain. Significant insights into the dynamics of the cassava value chain have been determined and directions on what is required for the further development of the value chain have been provided, particularly for policymakers and stakeholders. The findings suggest that gender influences cassava sales and positive income, whereby being female boosts the likelihood of selling cassava and earning more income. As a result, actions and policies promoting gender equality and empowering women involved in the cassava value chain are urgently required. These interventions can involve offering training and capacity-building programmes tailored exclusively to female farmers, allowing women to participate in decision-making processes, and guaranteeing fair access to resources and market possibilities. Access to output markets is another crucial aspect determining the likelihood of selling cassava and earning a profit. Findings show a significant positive association between market access and cassava sales and positive income generation. As a result, national and provincial governments should prioritize increasing market linkages, improving transportation infrastructure, availing agro-processing equipment and machinery, and providing cassava growers with market information and intelligence. These interventions are bound to help farmers access potential consumers, provide opportunities for value addition and product diversification, and eventually contribute to higher levels of income generation throughout the cassava value chain.

According to our findings, age has a substantial impact on cassava sales and the generation of a positive income. The age coefficient suggests that increasing in age by one year correlates to a drop in the likelihood of selling cassava and generating a positive income. This research highlights the difficulties older farmers encounter when selling cassava and shows that age-related factors may influence their participation in the cassava value chain. Cassava farming, which entails planting, harvesting, and processing operations, can be physically demanding for older farmers. Physical strength and endurance reduction commonly occurring with ageing can impair their ability to fully participate in the cassava

value chain. As a result, older farmers may reduce their involvement in cassava production, which can affect cassava supply along the value chain. Specific policies and interventions suited to the requirements of elderly farmers should be considered to address this issue. Such measures may include providing training and capacity-building programmes tailored to their specific needs, ensuring access to appropriate technologies and equipment that facilitate their participation in the value chain, and establishing supportive networks or cooperatives that allow them to effectively collaborate and share resources.

Another essential factor in fostering cassava sales and positive income is livestock ownership. The beneficial effect of animal ownership on the chance of selling cassava highlights the potential benefits of combining livestock rearing and cassava growing. Farmers can use cassava leftovers, such as cassava peels, as livestock feed, lowering feed costs and increasing profitability. Furthermore, using livestock manure as a fertilizer for cassava cultivation can result in enhanced productivity. Therefore, there is a need for policymakers and stakeholders to promote integrated farming methods by providing farmers with financial assistance and training, among other incentives, to produce cassava and livestock. This is bound to boost income diversification and strengthen farming households' resilience against eminent shocks such as climate variability and price volatility on major staple food items. At the same time, access to extension services and food harvesting characteristics did not reveal statistically significant effects on cassava sales and positive income in this study. Further research on the relationship between the studied variables should be undertaken based on more granular data. This is bound to foster more evidence-based interventions and the development of tailored policies through which any potential impediments or restrictions to cassava sales and income generation might be overcome. This study was limited due to a few challenges, including the high geographic heterogeneity of farmers across three provinces (KwaZulu-Natal, Mpumalanga, and Limpopo), which introduced variability that required more data variables in order to use other methods to obtain a substantial result regarding our objectives. Therefore, we used logistic regression; however, its simplicity did not allow us to capture complex and nonlinear data relationships. Therefore, further in-depth research using region-specific analyses and considering alternative modelling approaches would enhance the study's robustness and relevance. This will assist in the considerable progress of generating more practical implications that are needed to provide better reference or guidance pertaining to the cassava value chain in South Africa.

**Author Contributions:** Conceptualization, M.H.L. and B.M.; methodology, B.M. and M.H.L.; software, B.M.; validation, M.H.L., B.M. and B.Z.; formal analysis, B.M.; investigation, M.H.L., B.M., B.Z. and N.T.; resources, M.H.L.; data curation, B.M. and M.H.L.; writing—original draft preparation, B.M.; writing—review and editing, M.H.L., B.Z. and N.T.; supervision, M.H.L.; project administration, M.H.L.; funding acquisition, M.H.L. All authors have read and agreed to the published version of the manuscript.

**Funding:** This research was funded by the Technology Innovation Agency (TIA), an implementing agency of the Department of Science and Innovation (DSI) in South Africa, grant number 2021/FUN116/AA.

**Institutional Review Board Statement:** Not applicable.

**Informed Consent Statement:** Not applicable.

**Data Availability Statement:** Data are available on request.

**Acknowledgments:** The research team is grateful to FABCO, a farmers' primary cooperative based in Tzaneen (Limpopo province), and agricultural advisors from the Greater Letaba, uMhlathuze, and uMhlabuyalingana local municipalities for their involvement in the primary data collection exercise. The valuable input and comments by the anonymous reviewers and the editor, who assisted in improving this paper, are highly appreciated.

**Conflicts of Interest:** The authors declare no conflict of interest.

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
