# Peer review of "Factors Influencing Cassava Sales and Income Generation among Cassava Producers in South Africa"

_sustainability, doi:10.3390/su151914366_

Round 1

Reviewer 1 Report

The study examines the factors that influence cassava sales and income generation. The authors used logistic regression to examine the effects of explanatory variables on the probability of selling Cassava and achieving a positive income. I would like to add a few minor comments to the article:

  • Line 204 – logit model.
  • From the methodology, it is unclear what the authors will model and what exactly the explained variables are. Only in paragraph 3.2.1 do they state that they work with four models. Nevertheless, they report the results of two models (with the variable Sold Cassava and Positive Cassava Income).
  • There is no description (basic statistics) of the explained variables (Sold Cassava and Positive Cassava Income). So Positive Cassava Income is profit from the sale of Cassava? It is not stated how they measured the profit from the sale of Cassava and calculated the costs of cassava production in individual farms.
  • The authors should better describe the variables used in the model (Table 1 and related text). In particular, carefully state the units of measurement, e.g. for household size specify that it is the number of people (or an equivalence scale), specify the period for monetary values (monthly), and specify the area unit for yield. Correct Table 1 – Sold Cassava should be 0/1 instead Kgs.
  • What is the representation of farms that do not grow Cassava (Land size Cassava has a minimum of 0), and why are they included in the sample? Why are there farms in the sample that do not farm land (or do the data refer to owned land and these farm on leased land)?
  • Figure 2 is not self-explanatory; authors may consider removing it - replacing it with just a comment.

Author Response

Editors-in-Chief, Sustainability

Dear Prof. Dr. Marc A. Rosen,

attached, please find a response to the reviewer's comments for your consideration.

Regards

Moses

Reviewer 2 Report

The paper is interesting and well written. However, there are still some details that must be taken into account before publication.

First, I would like to see some cassava figures in order to understand the plant and its use. It is recommended to add some figures into the text for the readers also.

The paper is too brief as the literature review section is missing. Please add one describing previous and relative research to the field of cassava, and agriculture in general.

Lastly, you should quote more comments regarding the paper's contribution to the profitability and sustainability of cassava producers.  

Thank you! 

Author Response

Editors-in-Chief, Sustainability

Dear Prof. Dr. Marc A. Rosen,

Attached, please find a response to the reviewer's comments.

Regards

Moses

Reviewer 3 Report

Why was the Cassava Sales approach chosen in this paper? What are the main scientific questions answered by this research? These issues need clearer clarification. The theoretical analysis is still modest to fully reflect the inherent logical relationship between Cassava Selling and Income that affects it. Does the Cassava Sales problem in the case area have salient characteristics? Is the problem representative? These issues need clearer clarification in this method. A more in-depth analysis of the mechanism of Cassava Sales on a time left approach that affects Cassava Producers is needed in the analysis of the results and theoretical implications of the theoretical value of this research, and a more in-depth analysis and a more targeted analysis of the practical implications are needed to provide better reference or guidance for the case area. Some limitations of the study area analysis, logical theoretical relationships in South African Cassava Producers and Social Comparison, the basis of the selected plot and sample, and targeted suggestions should be more clearly illustrated.

Author Response

Editors-in-Chief, Sustainability

Dear Prof. Dr. Marc A. Rosen,

Herewith attached, please find a response to the reviewer's comments. 

Regards

Moses 

Round 2

Reviewer 2 Report

No comment

Reviewer 3 Report

The author has made revisions according to the instructions, so I approve this paper for publication.